# MEMORY RETAINING FINETUNING VIA DISTILLATION

## ABSTRACT

Large language models (LLMs) pretrained on large corpora of internet text possess much of the world knowledge. Following pretraining, one often needs to conduct continued pretraining on certain capabilities such as math and coding, or "posttraining" (a.k.a., alignment) techniques to make the models follow users' instructions and align them with human preferences. One challenge during these finetuning stages is that the model can lose the pretraining knowledge or forget certain capabilities (e.g., in-context learning ability). Moreover, although there exist strong open-weight LLMs such as Llama 3, both their pretraining and posttraining data are not open to the public, making it difficult to mix the finetuning data with the models' own pretraining data as a solution for mitigating forgetting. We propose *label annealing*, a method that mitigates forgetting during finetuning without requiring access to the original pretraining data. Label annealing distills pretraining knowledge during finetuing by adding a KL divergence term in the loss function, regularizing the divergence between the finetuned model's predictions to those of the initial pretrained model. In mathematics and code finetuning, label annealing improves the model's performance in target domains without sacrificing other capabilities of the pretrained model. In alignment finetuning, our method introduces a smooth tradeoff between the instruction-following capability and the pretraining knowledge. We complement our empirical investigation with a mathematical model with overparameterized linear regression that provides geometric intuition why label annealing would help.

## 1 INTRODUCTION

Language models pretrained on large volume of internet text possess much of the common world knowledge (Achiam et al., 2023; Team et al., 2023; Reid et al., 2024; Anthropic, 2024; Gunter et al., 2024; Touvron et al., 2023b; Dubey et al., 2024b; Yang et al., 2024a; Jiang et al., 2023). With the advancement of open-weight model, e.g., Llama series (Touvron et al., 2023a; Dubey et al., 2024a), an emerging paradigm is to finetune those open-weight models to adapt to a wide range of downstream applications such as mathematics (Lewkowycz et al., 2022; Azerbayev et al., 2024), programming (Rozière et al., 2024a), medicine (Yuan et al., 2024; Chen et al., 2023), and law (Colombo et al., 2024b;a), or to follow human instructions and preferences (Ouyang et al., 2022).

One major challenge in finetuning is that the model may forget some old knowledge or capabilities. A standard solution is to mix the finetuning data with the upstream training data of the model. This is known as "experience replay" in the continual learning literature (Lopez-Paz & Ranzato, 2017; Chaudhry et al., 2019). Unfortunately, the training data of powerful open-weights models, e.g., Llama series (Dubey et al., 2024a; Touvron et al., 2023a), is never shared with the public, making it impossible to mix custom finetuning data with the original training data. Moreover, the emerging trend of learning specific knowledge or capability using synthetic data (Yang et al., 2024b; Zelikman et al., 2022; 2024) and the heavy use of expensive human annotation (Touvron et al., 2023a; Dubey et al., 2024a; Lightman et al., 2024) make it even harder to apply the experience replay solution. These trends necessitate finetuning methods that preserve the model's old knowledge with only access to its weights.

A simple idea is to add weight decay toward the initial weights instead of zeros (Loshchilov & Hutter, 2019; Kumar et al., 2023), constraining the model weights from diverging too far from their initialization. We show that this existing wisdom provides limited mitigation in the setting of language modeling, due to its inability to take finetuning data into consideration. Instead, we

Figure 1: **Label annealing** keeps a copy of the initial model and freezes it. During finetuning, it computes two forward passes, one with the finetuning model, and the other with the initial model. Then it regularizes the usual finetuning objective with the KL divergence between the output of two networks, mitigating forgetting that arises from finetuning.

propose a *data-dependent* approach by distilling the knowledge (Hinton et al., 2015) from the initial model during the finetuining process. We refer to this approach as *label annealing*. Concretely, we load an independent copy of the initial model and keep it frozen during training. We then add a regularization term that penalizes the KL divergence between the predicted token probabilities of the finetuned model and those of the initial model (Figure 1).

To demonstrate the effectiveness of label annealing, we design four empirical finetuning tasks: mathematics finetuning, code finetuning (Section 3.2), supervised instruction finetuning, and niche domain knowledge finetuning (Section 3.3). In each task, we design a specific finetuning dataset and some "target benchmarks", those we wish to see improvement due to finetuning, as well as some "source benchmarks" that measures if the general capabilities are preserved. We show that label annealing can sometimes mitigate forgetting at no cost of sacrificing the target benchmarks performance, and in other cases, it introduces a smooth tradeoff between target and source benchmarks.

We complement our empirical findings with a theoretical analysis of label annealing using an over-parameterized linear regression model (Section 4). We analyze the gradient descent solution of three finetuning techniques: direct fine-tuning, $L_2$ regularization toward initialization, and label annealing (Theorem 1). We show that, direct finetuning discards the information contained in the initial weights that lies within the span of the finetuning data, therefore only preserving the initial knowledge outside that span. In contrast, label annealing introduces a regularization term that preserves the model's knowledge both inside and outside the span of the finetuning data.

To summarize, our key contributions are:

- We identify the emerging task of mitigating forgetting when finetuning open-weight language models with only access to the model weights, and propose **label annealing** as a simple solution.
- We provide empirical results across multiple domains, including math, coding, and alignment, demonstrating that label annealing improves performance in the target domains while maintaining or minimizing degradation in general capabilities.
- We offer a clear theoretical explanation for why label annealing is more effective than direct fine-tuning or $L_2$ regularization toward initialization, providing geometric intuitions for our findings.

Label annealing is straightforward to implement, making it an attractive option for finetuning open-weight language models. From a scientific perspective, label annealing demonstrates that a model's knowledge can be preserved during finetuning without access to the original training data.

## 1.1 RELATED WORK

**Continual learning and pretraining.** The field of continual learning emerged from early research in neural networks (McCloskey & Cohen, 1989; Ratcliff, 1990), which studied how systems could learn from sequentially presented information (Schlimmer & Fisher, 1986; Grossberg, 2012). A central challenge in this area is preventing "catastrophic forgetting" – where neural networks lose previously acquired capabilities when learning new tasks (Robins, 1995; French, 1999; Goodfellow et al., 2015; Kemker et al., 2018; Kirkpatrick et al., 2017). Researchers have developed various approaches to address this challenge, including: parameter regularization techniques to maintain critical network weights (Nguyen et al., 2017; Zenke et al., 2017; Kirkpatrick et al., 2017), methods that expand model architectures (Rusu et al., 2016; Golkar et al., 2019), and memory-based solu-

tions that retain examples from previous tasks (Rebuffi et al., 2017; Shin et al., 2017; Lopez-Paz & Ranzato, 2017). While traditional continual learning research has focused on clearly defined sequential tasks, our work examines the practical scenario of finetuning large-scale open-source language models (Dubey et al., 2024a; Jiang et al., 2023). In the context of language models, researchers have shown that additional pretraining can effectively adapt models to specialized domains like programming (Rozière et al., 2024b), healthcare (Chen et al., 2023), and mathematics (Lewkowycz et al., 2022; Shao et al., 2024; Azerbayev et al., 2024). This adaptation process has been refined through advances in causal language modeling techniques (Gupta et al., 2023; Ibrahim et al., 2024; Parmar et al., 2024). Some work has investigated using experience replay to preserve capabilities (Gupta et al., 2023; Parmar et al., 2024), but these studies have been limited to either small models (around 400M parameters) or settings with access to extensive computational resources for training from scratch.

**Knowledge distillation and its variants.** Knowledge distillation (Buciluundefined et al., 2006; Hinton et al., 2015) tackles the task of training a high-quality, small student model under the supervision of a large teacher model. In the context of language modeling, knowledge distillation is mainly used in a black-box manner, where the student model only have access to the tokens generated by the teacher model (Taori et al., 2023; Fu et al., 2023). With the advancement of open-weight language models, there is a growing interest in white-box distillation, where the student model have full access to the logits output of the teacher model (Wen et al., 2023; Gu et al., 2024; Agarwal et al., 2024). Our paper differs from the knowledge distillation literature in that our goal is not compressing a larger model to a smaller one. Instead, we use the technique of knowledge distillation to mitigate forgetting in the finetuning stage of language models.

In the context of knowledge distillation, similar to our work is the study of self-distillation (Zhang et al., 2019; Mobahi et al., 2020), where they find that first training a model and then distilling the model can be more effective than directly training the model. The goal of self-distillation is to improve generalization, as opposed to mitigate forgetting. Closely related to self-distillation is the technique of label smoothing (Szegedy et al., 2016; Müller et al., 2019b;a) that regularizes the KL divergence between model output and the uniform distribution. Label smoothing is similar to our proposal in that we both regularize the probability distribution predicted by the model with an additional distribution over class labels — a uniform distribution in the context of label smoothing and the distribution predicted by the initial model in our method.

## 2 OUR METHOD

Before the advancement of the large pretrained language model followed by instruction tuning (Brown et al., 2020), there is less practical concern about forgetting when finetuning a pretrained model. For example, in language modeling, when we finetune a BERT model (Devlin et al., 2019) to perform a question-answering (Rajpurkar et al., 2016) task, we do not have concern if the finetuned model forgets its ability to perform other tasks (e.g., summarization) due to the *task-specific* nature of finetuning. In contrast, when adapting an open-weight language model, e.g., Llama (Dubey et al., 2024a), Mistral (Jiang et al., 2023), to a new domain of knowledge, one needs to make sure that the finetuned model is a reasonable model that can, for example, follow user instructions or complete few-shot examples (MetaAI, 2024).

In this section, we describe our proposed label annealing algorithm that aims to address the forgetting issue when finetuning an open-weight language model. We first introduce the setup for our method (Section 2.1), and then describe the label annealing regularization in detail (Section 2.2). To provide background for the empirical experiments in Section 3, we present the label annealing algorithm in the context of finetuning a language model, but we note that our method can be generalized to any supervised or self-supervised learning task that uses cross-entropy loss.

### 2.1 SETUP

**Finetuning dataset.** In the context of language modeling, we use $x$ to denote a sequence of tokens and $y$ to denote the next token following $x$. We use $(x, y) \sim \mathcal{D}_{\text{FT}}$ to denote a context sampled from the finetuning dataset $\mathcal{D}_{\text{FT}}$. Let $p_{\theta}(y|x)$ denote the output distribution of our model, where $\theta$

represents the model weights. We denote the output distribution of the initial model as $p_{\boldsymbol{\theta}_0}(y|\boldsymbol{x})$, where $\boldsymbol{\theta}_0$ are the initial parameters.

**Success criteria.** We consider methods that only require access to the weights of the initial model. For each finetuning dataset, we define two types of evaluation benchmarks: (1) target benchmarks that measure improvement in the specific domain being finetuned, and (2) source benchmarks that assess preservation of the model's general capabilities. Then, we consider a technique as successfully mitigating the forgetting problem if (i) the improvement in the target benchmark is close to that of direct finetuning without applying any memory retaining technique, and (ii) the performance on the source benchmarks is close to that of the initial model.

## 2.2 LABEL ANNEALING

In this section, we introduce the label annealing regularization in the context of language model finetuning (Figure 1). Without any regularization, causal language modeling aims to minimize the following objective:

$$L(\boldsymbol{\theta}) = \mathbb{E}_{\boldsymbol{x},y\sim\mathcal{D}}[-\log p_{\boldsymbol{\theta}}(y|\boldsymbol{x})]. \tag{1}$$

However, aggressively optimizing this objective may cause the model to forget its old capabilities. To mitigate this, label annealing introduces a regularization term with temperature scaling:

$$L_{\text{LA}}(\boldsymbol{\theta}, T) = \lambda\mathbb{E}_{\boldsymbol{x},y\sim\mathcal{D}}[\mathsf{KL}(p_{\boldsymbol{\theta},T}(y|\boldsymbol{x})\|p_{\boldsymbol{\theta}_0,T}(y|\boldsymbol{x}))], \tag{2}$$

where $\mathsf{KL}$ denotes the Kullback-Leibler divergence, and $T$ is the temperature scaling parameter. The temperature-scaled distributions are defined as:

$$p_{\boldsymbol{\theta},T}(y|\boldsymbol{x}) = \frac{\exp(z_y/T)}{\sum_{y'}\exp(z_{y'}/T)},$$

where $\boldsymbol{z}$ represents the logits output of the model for input $\boldsymbol{x}$. The same scaling is applied to $p_{\boldsymbol{\theta}_0,T}(y|\boldsymbol{x})$ using the pretrained model's logits. The full objective then becomes:

$$L_{\text{total}}(\boldsymbol{\theta}) = L(\boldsymbol{\theta}) + L_{\text{LA}}(\boldsymbol{\theta}, T) \tag{3}$$

where $\lambda$ is a hyperparameter controlling the strength of the regularization, and $T$ controls the "sharpness" of the distributions. A higher temperature ($T > 1$) makes the distributions more uniform, while a lower temperature ($0 < T < 1$) makes them more peaked. In the limit of $T \to \infty$, our regularization reduces to the label smoothing (Szegedy et al., 2016) regularization discussed in Section 1.1. By optimizing this combined objective, label annealing seeks to strike a balance between adapting to the new data and retaining knowledge from pretraining.

## 3 MAIN EXPERIMENTS

In this section, we describe our main experiments. We finetune Llama 3 8B base model (Dubey et al., 2024a) on various downstream datasets, and measure its performance on some standard language model benchmarks. We split our experiments into two sections.

In Section 3.2, we finetune the model on additional mathematics and code data. We find that label annealing prevents forgetting with no loss in downstream performance. In contrast, a simple baseline of adding $L_2$ regularization toward initialization (Kumar et al., 2023) provides little to no benefit. In Section 3.3, we present two complementary experiments, (i) performing instruction tuning on the base model and (ii) finetuning an aligned model on an additional downstream dataset. In both cases, we observe a smooth tradeoff between finetuning benchmarks and pretraining benchmarks. We next present the experiment setup that serves this section.

## 3.1 EXPERIMENT SETUP

**Evaluation setup.** Each finetuning dataset $\mathcal{D}_{\text{FT}}$ is evaluated using two benchmark categories: (1) target benchmarks $\mathcal{B}_{\text{target}}$ that measure improvement in finetuned capabilities, and (2) source benchmarks $\mathcal{B}_{\text{source}}$ that verify preservation of the original model's performance. As an example, if we perform additional training on mathematics related text, we would expect math benchmarks like GSM8K (Cobbe et al., 2021) or MATH (Hendrycks et al., 2021b) to improve, whereas general capability benchmarks such as MMLU (Hendrycks et al., 2021a) to not degrade.

**Benchmark selection.** We select 8 benchmarks, roughly divided into five categories, to probe different knowledge or capability of the finetuned model. Note that we do not evaluate each finetuning experiment with data source $\mathcal{D}_{FT}$ on all 8 benchmarks. Instead, based on the content of each finetuning dataset, we select $\mathcal{B}_{target}$ as the benchmarks we expect to see improvement and select $\mathcal{B}_{source}$ as the benchmarks we wish to retain performance. We defer the evaluation details to Appendix A. At a high level, we pick benchmarks that probe performance in mathematics (MATH (Hendrycks et al., 2021b), GSM8K (Cobbe et al., 2021)), coding (HumanEval (Chen et al., 2021)), pretraining knowledge (MMLU (Hendrycks et al., 2021a), TriviaQA (Joshi et al., 2017)), niche books and articles (QuALITY QA (Pang et al., 2022; Yang et al., 2024b)).

**Direct finetuning baseline.** For each finetuning dataset $\mathcal{D}_{FT}$, we first report the result of directly finetuning on it. As we shall see, in some cases the forgetting is not a serious problem even without any intervention. Throughout our experiments, we use batch size 128 and context window 2048, leading to a throughput of 262K tokens per batch. We use cosine learning rate decay with linear warmup up to 5% of total steps with peak learning rate 5e-6. For all datasets, we finetune on the downstream dataset for 5 epochs. As a result, we standardize a fixed set of training hyperparameter and do not perform dataset specific hyperparameter selection for each finetuning dataset $\mathcal{D}_{FT}$.

$L_2$ **regularization baseline.** A simple baseline to maintain pretraining knowledge with only access to pretrained weights is to simply regularize the weights toward initialization. We consider this simple approach of $L_2$ regularization as a baseline. This is introduced as "regenerative regularization" in Kumar et al. (2023), it can also be viewed as a simplified case of Elastic Weight Consolidation (EWC) (Kirkpatrick et al., 2017), where all parameters are weighted equally. We implement it by adding the penalty term $\frac{\lambda}{2}\|\boldsymbol{\theta} - \boldsymbol{\theta}_0\|_2^2$ to the primary objective (1). We follow the same set of training hyperparameter as in direct finetuning. To select the best $\lambda$, we sweep over a range of $\lambda$ for each finetuning dataset $\mathcal{D}_{FT}$. Then, we first filter out those choice of $\lambda$ that lead to no improvement in target benchmarks $\mathcal{B}_{target}$, and then select the one that has the highest value in source benchmark $\mathcal{B}_{source}$.

**Label annealing setup.** For our approach, we follow the same hyperparameter choice for training as in direct finetuning and $L_2$ regularization. To select the label annealing hyperparameter, namely, the annealing scale $\lambda$ and temperature $T$ from (2), we follow the same selection process as in $L_2$ regularization: pre-filtering based on target benchmark $\mathcal{B}_{target}$ performance, and then select the one with best source benchmark performance $\mathcal{B}_{source}$.

### 3.2 Base model training

In this section, we describe experiments where we finetune the Llama 3 8B model with two finetuning datasets $\mathcal{D}_{FT}$. We describe the dataset construction below, as well as the target benchmarks $\mathcal{B}_{target}$ and source benchmarks $\mathcal{B}_{source}$ for each dataset.

**Mathematics finetuning.** We first perform finetuning on some synthetic mathematics corpus. Our dataset $\mathcal{D}_{FT}$ is constructed in two stages. (i) We start with a seed dataset of math QA dataset from Metamath (Yu et al., 2024) and additional questions collected from StackExchange. (ii) We generate a synthetic corpus by prompting GPT-4o-mini (OpenAI et al., 2024) to convert the question and answer pairs to a textbook-like article (Full prompts in Appendix A.2). Following the steps above, we generate a corpus with 179M tokens.

Training on the math-related corpus, we expect mathematics benchmarks to improve $\mathcal{B}_{target} = \{MATH, GSM8K\}$ and we hope to maintain the same performance in pretraining metrics $\mathcal{B}_{source} = \{HumanEval, MMLU, TriviaQA\}$. Indeed, we see in Table 1 that direct finetuning improves mathematics performance at the cost of hurting pretraining metrics, particular so in TriviaQA (drop by 14.19%), which tests more tail knowledge as opposed to commonsense knowledge. While the $L_2$ regularization failed to resolve the forgetting problem, label annealing fixes the forgetting in pretraining metrics (MMLU and TriviaQA) and in some cases improving performance in target benchmarks. Note that even though our synthetic mathematics corpus $\mathcal{D}_{FT}$ does not directly aim to improve coding ability, direct finetuning on this corpus does improve HumanEval performance, and label annealing can preserve most of the improvement.

| Training recipe | Mathematics | | Coding | Pretraining | |
|---|---|---|---|---|---|
| | MATH | GSM8K | HumanEval | MMLU | TriviaQA |
| Llama 3 8B Base | 15.92 | 51.17 | 28.77 | 65.03 | 67.99 |
| Direct finetuning | 17.10 | 62.01 | 38.31 | 62.54 | 53.80 |
| $L_2$ regularization | 16.78 | 62.24 | 38.32 | 62.24 | 53.51 |
| Label annealing | 17.94 | 61.78 | 35.24 | 64.62 | 65.87 |

Table 1: Mathematics finetuning results. Target benchmarks $\mathcal{B}_{\text{target}}$={GSM8K, MATH} and source benchmarks $\mathcal{B}_{\text{source}}$ ={HumanEval, MMLU, TriviaQA}. Label annealing resolves the forgetting issue introduced in direct finetuning, while $L_2$ regularization provides little help.

**Code finetuning.** Another common finetuning task is code-specific finetuning (Rozière et al., 2024b). Base pretrained language models like Llama 3 Base are typically already trained on a large amount of code data. This process gives them some basic code completion ability. One typical approach to enable the model to perform more complex coding tasks is via code-specific instruction tuning. Toward this goal, we adopt a code instruction corpus constructed based on the method in StarCoder (Li et al., 2023a) as our finetuning dataset $\mathcal{D}_{\text{FT}}$ of 30M tokens.

| Training recipe | Coding | Mathematics | | Pretraining | |
|---|---|---|---|---|---|
| | HumanEval | MATH | GSM8K | MMLU | TriviaQA |
| Llama 3 8B Base | 28.77 | 15.92 | 51.17 | 65.03 | 67.99 |
| Direct finetuning | 54.53 | 1.19 | 37.07 | 64.82 | 64.60 |
| $L_2$ regularization | 53.00 | 11.36 | 34.87 | 64.87 | 64.48 |
| Label annealing | 51.06 | 17.16 | 52.69 | 64.63 | 67.24 |

Table 2: Code finetuning results. Target benchmark $\mathcal{B}_{\text{target}}$ ={HumanEval} and source benchmarks $\mathcal{B}_{\text{source}}$ ={MATH, GSM8K, MMLU, TriviaQA}. Label annealing resolves the forgetting in mathematics benchmarks while also preserving the most of the improvement in HumanEval.

Training on this corpus, we expect to see improvement on code metrics $\mathcal{B}_{\text{target}}$ ={HumanEval} and preserve other metrics $\mathcal{B}_{\text{source}}$ ={MATH, GSM8K, MMLU, TriviaQA}. Indeed, we see in Table 2 that the HumanEval performance improves by 25.76% with some degradation in mathematics metrics (MATH and GSM8K). Note that we see a dramatic drop in the MATH benchmark (drop by 14.73%). Reading into the generated samples, we find the model sometimes loses its ability to follow few-shot prompts. $L_2$ regularization is able to resolve this issue, but still losing performance in both MATH and GSM8K. In contrast, label annealing is able to resolve the forgetting problem, and also preserve most of the improvement in HumanEval.

### 3.3 ALIGNED MODEL TRAINING

In this section, we present experiments involving instruction-tuned language model. We consider two complementary scenarios, (i) perform instruction tuning on a pretrained language model and see if we can get instruction-following ability without compromising the pretraining metrics, a problem known as "alignment tax", and (ii) perform knowledge intensive finetuning on an aligned model (e.g., Llama 3 8B Instruct) and see if we can "teach" model the new knowledge while preserving the instruction following ability. In both cases, we observe that label annealing introduces a smooth tradeoff between the target benchmarks $\mathcal{B}_{\text{target}}$ and source benchmarks $\mathcal{B}_{\text{source}}$.

**Alignment tax.** The standard step following pretraining is instruction tuning (Ouyang et al., 2022). Models are known to lose some of their pretraining knowledge during the instruction tuning stage (e.g., MMLU drop in Llama 3 (Dubey et al., 2024a)). To investigate forgetting in this setting, we perform supervised instruction tuning on the UltraChat (Ding et al., 2023) dataset. After tokenizing it with the standard Llama 3 Instruct template, we get an instruction tuning corpus with 220M tokens.

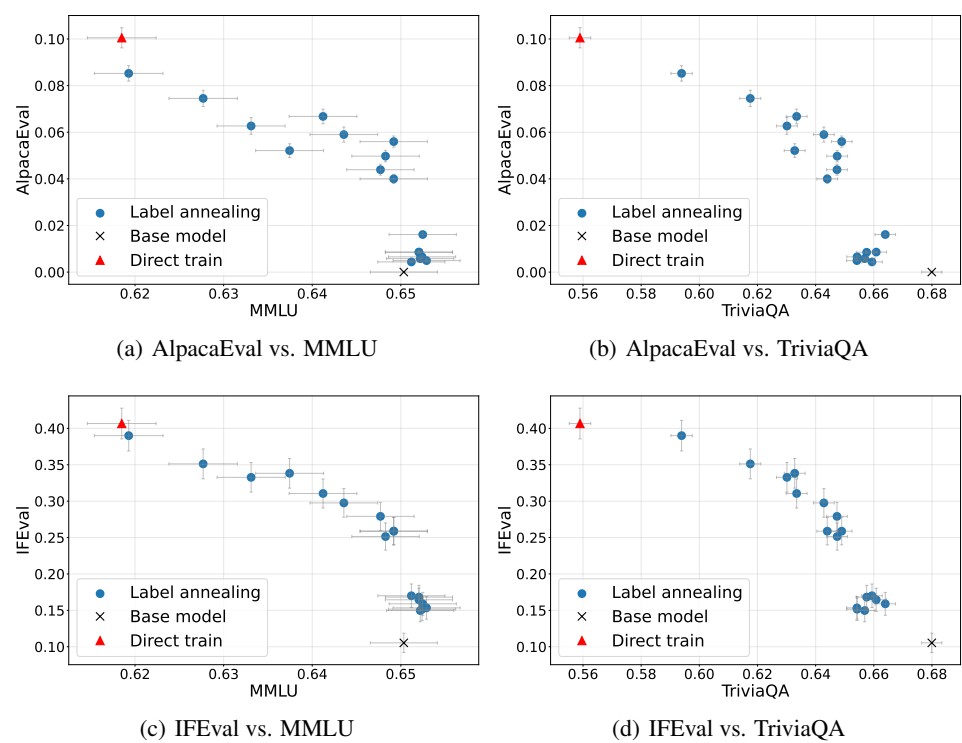

(a) AlpacaEval vs. MMLU

(b) AlpacaEval vs. TriviaQA

(c) IFEval vs. MMLU

(d) IFEval vs. TriviaQA

Figure 2: Instruction tuning on Llama 3 8B. **(y-axis)** Target benchmarks $\mathcal{B}_{\text{target}}$ ={AlpacaEval, IFEval} v.s. **(x-axis)** source benchmarks $\mathcal{B}_{\text{source}}$ ={MMLU, TriviaQA}. Label annealing with different magnitude offers a smooth tradeoff between $\mathcal{B}_{\text{target}}$ and $\mathcal{B}_{\text{source}}$.

We expect training on this dataset to improve the instruction following ability, as measured by $\mathcal{B}_{\text{target}}$ ={AlapcaEval, IFEval}. We select source benchmarks $\mathcal{B}_{\text{source}}$ ={MMLU, TriviaQA} that measure pretraining quality of the base model. In Figure 2, we plot target metrics against source metrics for label annealing with different value of $\lambda$ in (2). We can see that label annealing introduces a smooth tradeoff between pretraining metrics and instruction following metrics. In some cases, for example Figure 2(a) and 2(c), we can get about half of improvement in instruction following ability without loosing MMLU knowledge.

**Niche books and articles QA.** Finally, consider the task of continually pretrain an instruction-tuned model. Typically, instruction tuning is the last step of building a large language model, so no training should happen beyond that point. However, with the release of powerful open-source instruction-tuned models, one might consider taking a model like Llama 3 8B Instruct and finetune it to a niche domain whose knowledge rarely appears during the pretraining phase. We investigate the performance of the label annealing method in this setting.

To design experiments, we find that the benchmarks we have looked are overly saturated for Llama 3 8B Instruct. For example, Llama 3 8B Instruct has MATH performance 51.9% and HumanEval 72.6% (Dubey et al., 2024a), making it difficult to measure improvement on top of this model. Instead of looking at naturally occurring benchmarks, we consider the QuALITY dataset introduced by Pang et al. (2022), which includes 4,609 reading comprehension questions about a collection of obscure books and articles as introduced in Section 3.1. Yang et al. (2024b) introduces a corpus of 455M tokens that are synthetic data related to the QuALITY articles. They find that training on this dataset greatly improves the QuALITY QA accuracy. This gives a natural pair of finetuning dataset $\mathcal{D}_{\text{FT}}$ (the 455M synthetic corpus) and target benchmarks $\mathcal{B}_{\text{target}}$ (the QuALITY QA accuracy) for our task. We finetune Llama 3 8B Instruct on this dataset using the same hyperparameter setup as in the base model finetuning (Section 3.2).

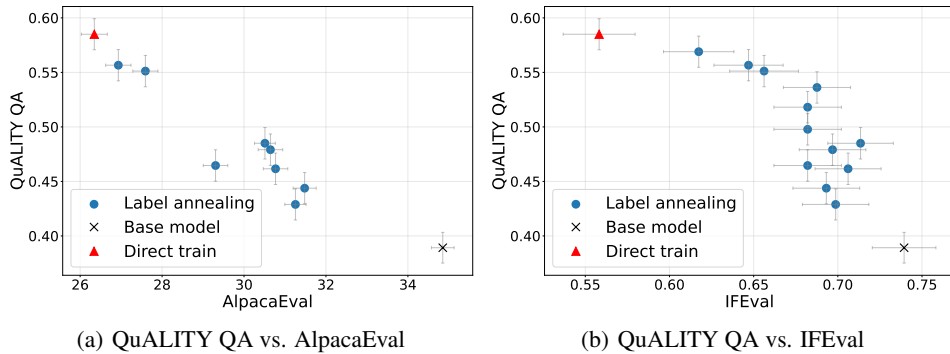

(a) QuALITY QA vs. AlpacaEval  (b) QuALITY QA vs. IFEval

Figure 3: Niche books and articles QA. Target benchmarks $\mathcal{B}_{\text{target}}$ ={Reading comprehension questions about the articles}. Source benchmark $\mathcal{B}_{\text{source}}$ ={AlpacaEval, IFEval}. Each dot correspond to one label annealing experiment with different magnitude.

We report our result in Figure 3. We can see that label annealing with different magnitude introduces a tradeoff between target benchmark $\mathcal{B}_{\text{target}}$ and source benchmarks $\mathcal{B}_{\text{source}}$. In Figure 3(b), we see that with some choice of label annealing hyperparameter, we can get more than 80% of improvement in QuALITY question set with a small reduction in IFEval metrics.

## 4 LABEL ANNEALING IN LINEAR REGRESSION

In this section, we analyze label annealing in the simple setting of over-parameterized linear regression. Concretely, we consider the task of first pretrain a linear model on one dataset and finetune on another. As in Section 3, we consider three strategies of finetuning: direct finetuning, $L_2$ regularization, and label annealing. We will provide a geometric intuition of these three different strategies.

### 4.1 PRETRAINING STEP

Let $\tilde{\boldsymbol{X}} \in \mathbb{R}^{N \times d}, \tilde{\boldsymbol{y}} \in \mathbb{R}^N$ be the covariate matrix and response vector of a linear regression task. We consider the overparameterized regime where $d > N$. We use the superscript $\sim$ to denote the variables related to the pretraining data. In the pretraining step, we solve the following optimization problem:

$$\textbf{Pretraining step: } \min_{\boldsymbol{\theta} \in \mathbb{R}^d} \frac{1}{2} \|\tilde{\boldsymbol{X}}\boldsymbol{\theta} - \tilde{\boldsymbol{y}}\|_2^2 \tag{4}$$

Note that even though the objective above is convex, it is not strictly convex because of the over-parameterized nature of the task. In fact, the global optimal is given by the affine subspace $\{\boldsymbol{\theta} \in \mathbb{R}^d : \tilde{\boldsymbol{X}}\boldsymbol{\theta} = \tilde{\boldsymbol{y}}\}$. As a result, the minimizer is implicitly selected by the initialization of the gradient descent algorithm, as characterized by the following proposition:

**Proposition 4.1** (Pretraining solution). *Suppose that $\tilde{\boldsymbol{X}}$ has strictly positive singular values and there exists $\boldsymbol{\theta}$ such that $\tilde{\boldsymbol{X}}\boldsymbol{\theta} = \boldsymbol{y}$. Then, the gradient descent applied to problem (6) with a learning rate less than $\sigma_{\max}^{-2}$ converges, where $\sigma_{\max}$ is the maximum singular value of $\tilde{\boldsymbol{X}}$. Moreover, if the gradient descent is initialized at $\boldsymbol{0} \in \mathbb{R}^d$, it will converge to a particular global optimal $\boldsymbol{\theta}_0 = \tilde{\boldsymbol{X}}^\top (\tilde{\boldsymbol{X}}\tilde{\boldsymbol{X}}^\top)^{-1}\tilde{\boldsymbol{y}}$.*

The proof follows from a direct application of Lemma B.1, which we prove in Appendix B. We denote the pretrained weights by $\boldsymbol{\theta}_0$, and

$$\boldsymbol{\theta}_0 = \tilde{\boldsymbol{X}}^\top (\tilde{\boldsymbol{X}}\tilde{\boldsymbol{X}}^\top)^{-1}\tilde{\boldsymbol{y}}. \tag{5}$$

In words, our modeling of the pretraining step assumes that the pretrained weights memorize the pretraining data with minimum Euclidean norm.

## 4.2 FINE-TUNING STEP

Given the pretrained weights $\boldsymbol{\theta}_0$, let $\boldsymbol{X} \in \mathbb{R}^{n \times d}, \boldsymbol{y} \in \mathbb{R}^n$ be a new dataset we would like to finetune on. As before, we assume that we are in the overparameterized regime with $d > n$. We will consider three finetuning approaches, the first approach is to directly train on $\boldsymbol{X}$ and $\boldsymbol{y}$:

$$\textbf{Direct tuning:} \quad \min_{\boldsymbol{\theta} \in \mathbb{R}^d} \frac{1}{2} \|\boldsymbol{X}\boldsymbol{\theta} - \boldsymbol{y}\|_2^2. \tag{6}$$

Next, as in Section 3, we consider finetuning with $L_2$ regularization (weight decay to initialization):

$$L_2 \textbf{ regularization:} \quad \min_{\boldsymbol{\theta} \in \mathbb{R}^d} \frac{1}{2} \|\boldsymbol{X}\boldsymbol{\theta} - \boldsymbol{y}\|_2^2 + \frac{\lambda}{2} \|\boldsymbol{\theta} - \boldsymbol{\theta}_0\|_2^2. \tag{7}$$

Finally, we claim that label annealing regularization in the context of linear regression corresponds to the objective:

$$\textbf{Label annealing:} \quad \min_{\boldsymbol{\theta} \in \mathbb{R}^d} \frac{1}{2} \|\boldsymbol{X}\boldsymbol{\theta} - \boldsymbol{y}\|_2^2 + \frac{\lambda}{2} \|\boldsymbol{X}\boldsymbol{\theta} - \boldsymbol{X}\boldsymbol{\theta}_0\|_2^2. \tag{8}$$

To see this, recall that in the language modeling setting, label annealing adds a KL divergence penalty between the logits from the current model and the logits from the pretrained model on each batch of finetuning data. From the perspective of neural network, the logits from the pretrained model means taking a forward pass with on the finetuning data with pretrained weights. If we simplify the transformer neural network to a single linear layer with pretrained weights $\boldsymbol{\theta}_0$, then the forward pass on the finetuning data $\boldsymbol{X}$ becomes $\boldsymbol{X} \to \boldsymbol{X}\boldsymbol{\theta}_0$. If we again simplify the KL divergence penalty with $L_2$ loss, we obtain the objective (8).

We next consider the finetuning process, where we initialize our linear weights as pretrained weights $\boldsymbol{\theta}_0$, as with a real transformer neural network. We characterize the solution gradient descent converges to for each of three finetuning approaches: direct tuning (6), $L_2$ regularization (7), and label annealing (8).

**Theorem 1** (Finetuning step.). *Suppose that $\boldsymbol{X}$ has strictly positive singular values and there exists $\boldsymbol{\theta}$ such that $\boldsymbol{X}\boldsymbol{\theta} = \boldsymbol{y}$. For $\lambda > 0$, gradient descent applied to objectives (6), (7), (8) with learning rate less than $\min\{\sigma_{\max}^{-2}, (\lambda + \sigma_{\max}^2)^{-1}, (\lambda\sigma_{\max}^2 + \sigma_{\max}^2)^{-1}\}$ converges. Moreover, if gradient descent is initialized at the pretrained weights $\boldsymbol{\theta}_0$, they converge to the following solution*

$$\textit{Direct tuning:} \quad \boldsymbol{\theta}_{Direct} = \left[\boldsymbol{I} - \boldsymbol{X}^{\mathsf{T}}(\boldsymbol{X}\boldsymbol{X}^{\mathsf{T}})^{-1}\boldsymbol{X}\right]\boldsymbol{\theta}_0 + \boldsymbol{X}^{\mathsf{T}}(\boldsymbol{X}\boldsymbol{X})^{-1}\boldsymbol{y},$$

$$L_2 \textit{ regularization:} \quad \boldsymbol{\theta}_{L2} = (\boldsymbol{X}^{\mathsf{T}}\boldsymbol{X} + \lambda\boldsymbol{I})^{-1}\boldsymbol{X}^{\mathsf{T}}\boldsymbol{y} + \lambda(\boldsymbol{X}^{\mathsf{T}}\boldsymbol{X} + \lambda\boldsymbol{I})^{-1}\boldsymbol{\theta}_0,$$

$$\textit{Label annealing:} \quad \boldsymbol{\theta}_{LA} = \left[\boldsymbol{I} - (1+\lambda)^{-1}\boldsymbol{X}^{\mathsf{T}}(\boldsymbol{X}\boldsymbol{X}^{\mathsf{T}})^{-1}\boldsymbol{X}\right]\boldsymbol{\theta}_0 + (1+\lambda)^{-1}\boldsymbol{X}^{\mathsf{T}}(\boldsymbol{X}\boldsymbol{X})^{-1}\boldsymbol{y}.$$

The proof of the theorem is another direct application of Lemma B.1, whose proof we defer to Appendix A. As a sanity check, we can see $\boldsymbol{\theta}_{\text{Direct}}$ can be viewed as two limiting cases of $\boldsymbol{\theta}_{L2}$ and $\boldsymbol{\theta}_{\text{LA}}$. For example $\lambda = 0$, we have $\boldsymbol{\theta}_{\text{LA}} = \boldsymbol{\theta}_{\text{Direct}}$. Alternatively, as $\lambda \to 0$, we have $\boldsymbol{\theta}_{L2} \to \boldsymbol{\theta}_{\text{Direct}}$.

## 4.3 INTERPRETATION OF THE RESULTS.

The solution selected by gradient descent admits straightforward geometric interpretation. Since we are in the overparametrized regime with $d > n$, the rows of $\boldsymbol{X}$ span a $n$-dimensional space, which intuitively corresponds to a subspace spanned by the finetuning data. The matrix $\boldsymbol{X}^{\mathsf{T}}(\boldsymbol{X}\boldsymbol{X}^{\mathsf{T}})^{-1}\boldsymbol{X}$ is then a projection onto this space. This space spanned by the finetuning data will be a core theme of this section.

**Direct finetuning.** The direct finetuning solution $\boldsymbol{\theta}_{\text{Direct}}$ consists of two components: $\left[\boldsymbol{I} - \boldsymbol{X}^{\mathsf{T}}(\boldsymbol{X}\boldsymbol{X}^{\mathsf{T}})^{-1}\boldsymbol{X}\right]\boldsymbol{\theta}_0$ is the projection of pretrained weights onto the orthogonal complement of the space spanned by the finetuning data, and $\boldsymbol{X}^{\mathsf{T}}(\boldsymbol{X}\boldsymbol{X})^{-1}\boldsymbol{y}$ is the minimum Euclidean norm solution, (Hastie et al., 2022; Bartlett et al., 2020) which is orthogonal to the first component. In words, direct finetuning would keep the portion of pretrained weights $\boldsymbol{\theta}_0$ outside the span of finetuning data $\boldsymbol{X}$ fixed, and ignore the information about $\boldsymbol{\theta}_0$ within the span of $\boldsymbol{X}$. Since $\boldsymbol{\theta}_0$ is necessarily in the span of pretraining data $\tilde{\boldsymbol{X}}$, our toy theory suggests that direct finetuning would avoid forgetting issue if the finetuning data and pretraining data are completely orthogonal.

$L_2$ **regularization.** The solution selected by $L_2$ regularization also has two terms, but they are no longer orthogonal as in direct finetuning. The first term $(\boldsymbol{X}^\mathsf{T}\boldsymbol{X} + \lambda\boldsymbol{I})^{-1}\boldsymbol{X}^\mathsf{T}y$ is the usual ridge regression solution with ridge penalty $\lambda$. The second term $\lambda(\boldsymbol{X}^\mathsf{T}\boldsymbol{X} + \lambda\boldsymbol{I})^{-1}\boldsymbol{\theta}_0$ "rescales" the pretrained weights $\boldsymbol{\theta}_0$ based on finetuning data $\boldsymbol{X}$. When $\lambda \to \infty$, this second term becomes exactly $\boldsymbol{\theta}_0$. As the two terms overlap with each other, we see that $L_2$ regularization admits no clean intuition why it would help. This is consistent with the empirical experiments (Section 3) that they tend to perform poorly compared with label annealing.

**Label annealing.** The label annealing solution $\boldsymbol{\theta}_{\mathrm{LA}}$ is a smoothed version of the direct finetuning solution. With the introduction of $\lambda$, $\left[\boldsymbol{I} - (1+\lambda)^{-1}\boldsymbol{X}^\mathsf{T}(\boldsymbol{X}\boldsymbol{X}^\mathsf{T})^{-1}\boldsymbol{X}\right]\boldsymbol{\theta}_0$ is no longer a projection onto the complement of spanned by the finetuning data. Instead, it adds back a small component along the direction $\left[\boldsymbol{I} - \boldsymbol{X}^\mathsf{T}(\boldsymbol{X}\boldsymbol{X}^\mathsf{T})^{-1}\boldsymbol{X}\right]\boldsymbol{\theta}_0$ scaled by $\lambda/(1+\lambda)$. Concretely, label annealing solution can be rewritten as

$$
\boldsymbol{\theta}_{\mathrm{LA}} = \overbrace{\left[\boldsymbol{I} - \boldsymbol{X}^\mathsf{T}(\boldsymbol{X}\boldsymbol{X}^\mathsf{T})^{-1}\boldsymbol{X}\right]\boldsymbol{\theta}_0}^{\text{Component orthogonal to the space spanned by finetuing data}}
$$
$$
+ \underbrace{\frac{\lambda}{1+\lambda}\boldsymbol{X}^\mathsf{T}(\boldsymbol{X}\boldsymbol{X}^\mathsf{T})^{-1}\boldsymbol{X}\boldsymbol{\theta}_0 + \frac{1}{1+\lambda}\boldsymbol{X}^\mathsf{T}(\boldsymbol{X}\boldsymbol{X})^{-1}\boldsymbol{y}}_{\text{Convex combination of pretrained weights in the finetuning data span and the minmum norm solution}}
$$

In some sense, label annealing is getting the best of both worlds — preserving the pretrained weights in the orthogonal complement of the space spanned by the finetuning data and also tradeoff between pretraining and finetuning information within the span of finetuning data $\boldsymbol{X}$.

## 5 LIMITATIONS

Despite the lack of open-soured dataset available for direct download, more or less some information about the training data of open-weight models can be found. For example, Bommasani et al. (2024) evaluated the "transparency index" of training data for Llama 2, GPT-4, Claude 3 as 40%, 20%, 0%, respectively. Based on the publicly available information, the RedPajama corpus (TogetherAI, 2023) is an effort to reconstruct the training data for Llama series of models. We report the performance of adding 10% replay from RedPajama corpus on the mathematics finetuing task (same setup as Section 3.2) below:

| Training recipe | Mathematics | | Coding | Pretraining | |
|---|---|---|---|---|---|
| | MATH | GSM8K | HumanEval | MMLU | TriviaQA |
| Llama 3 8B Base | 15.92 | 51.17 | 28.77 | 65.03 | 67.99 |
| Direct finetuning | 17.10 | 62.01 | 38.31 | 62.54 | 53.80 |
| Replay | 22.40 | 69.52 | 29.64 | 63.79 | 64.96 |
| Replay + label annealing | 23.44 | 69.21 | 31.72 | 64.05 | 65.07 |

Table 3: Math continued pretraining with replay on RedPajama.

We can see that adding replay from the RedPajama corpus happens to alleviate the forgetting issue in pretraining metrics (MMLU, TriviaQA) to some extent, performing on par with using replay and label annealing simultaneously. However, as the field moves toward more complex training strategies with synthetic data and proprietary data, it becomes increasingly difficult to reconstruct a training data that covers all the capabilities that would go beyond the coverage of MMLU and TriviaQA. In contrast, label annealing stands as a reliable method that mitigates forgetting during finetuning requiring only access to the weights of the finetuned model.

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

APPENDIX

# A EXPERIMENT DETAILS

## A.1 BENCHMARK SELECTION AND EVALUATION DETAILS

In this section, we present the details of benchmark we choose and evaluation strategy.

- Pretraining knowledge: MMLU (Hendrycks et al., 2021a) and TriviaQA (Joshi et al., 2017). Both benchmarks measure general world knowledge and typically used to measure model's pretraining knowledge. We view MMLU as a measurement of more common knowledge and TriviaQA as niche tail knowledge. Indeed, we will see that TriviaQA is more prone to forgetting compared with MMLU. We use 5-shot prompting for MMLU and 0-shot for TriviaQA.
- Mathematics ability: MATH (Hendrycks et al., 2021b) and GSM8K (Cobbe et al., 2021). Both benchmarks measure math problem-solving ability. MATH contains harder questions than GSM8K, so typically has lower accuracy and also harder to improve. We use 4-shot chain-of-thought Minerva prompt for MATH (Lewkowycz et al., 2022) and 4-shot prompting for GSM8K.
- Coding ability: HumanEval (Chen et al., 2021), an open-ended programming task. We report pass@1 performance for HumanEval.
- Instruction following ability: AlpacaEval (Dubois et al., 2024; Li et al., 2023b) and IFEval (Zhou et al., 2023). In our setup, we compute the AlpacaEval win rate against GPT-4 (OpenAI et al., 2024) using GPT-4 as judge.
- Niche articles and books knowledge: contextualized QuALITY QA (Pang et al., 2022; Yang et al., 2024b). The QuALITY dataset was originally proposed as a long-context reading comprehension task about a collection of 265 relatively obscure books and articles. In its original form, some questions such as "What does the author think?" are ambiguous without the article available as context. Yang et al. (2024b) turned the question set into a collection of unambiguous questions by appending the article metadata as "In the article {article title} by {author name}, what does the author think?". As a result, the article becomes a collection of unambiguous QA the probe model's knowledge about the collection of 265 books.

## A.2 MATHEMATICS SYNTHETIC DATA PROMPTS

```
## Instruction
Given the following question and answer data, please convert it into a
comprehensive educational text. Follow these steps:

* Start your response by explaining the key concepts and mathematical
principles involved. This can be algebra, geometry, probability or any
concepts and definitions that forms the background of the question.
* Use the provided question and answer as an exercise for
demonstrating how the provided concepts are used. State the questions
statement clearly.
* Provide a step-by-step solution, highlighting the critical thinking
process.
* Summarize the main learning points from this problem. What are the
key insights? What technique did you use to solve the problem?
* Discuss any broader mathematical concepts or applications related to
this problem.

Your goal is to create an educational text similar to a textbook that
not only solves the problem but also enhances the reader's
understanding of the underlying mathematical concepts. Be thorough in
your explanations while maintaining clarity. Your response should look
like a page of  a mathematics textbook.
```

### A.3 SYSTEM-LEVEL CONSIDERATIONS

Naively implementing the KL divergence and evaluate gradient using auto-grad function such as in PyTorch can lead to very unstable gradient. In this section, we manually run the backpropagation procedure up to the logits output $z = f_{\boldsymbol{\theta}}(\boldsymbol{x})$, and use this closed-from gradient as our implementation of label annealing algorithm. For notational simplicity, we denote the logits of the pretrained model as $\boldsymbol{w} = f_{\boldsymbol{\theta}_0}(\boldsymbol{x})$. The penalty incurred by the label annealing term is

$$l_{\text{LA}} = \mathsf{KL}[p_{\boldsymbol{\theta}_0,T}(\cdot|x)\|p_{\boldsymbol{\theta},T}(\cdot|x)] = \sum_y \frac{\exp(w_y/T)}{\sum_j \exp(w_j/T)} \log \left[ \frac{\frac{\exp(w_y/T)}{\sum_j \exp(w_j/T)}}{\frac{\exp(z_y/T)}{\sum_i \exp(z_i/T)}} \right].$$

Directly running auto-diff would result in numerical instability due to the large vocabulary size of a modern language model. If we manually evaluate the gradient, we find that the gradient of $l_{LA}$ is

$$\nabla_{\boldsymbol{z}} l_{\text{LA}} = \frac{1}{T} \left[ p_{\boldsymbol{\theta},T}(\cdot|x) - p_{\boldsymbol{\theta}_0,T}(\cdot|x) \right],$$

which can be stably calculated.

## B PROOF OF SECTION 4

We start by stating the following lemma, which analyzes the gradient solution of a quadratic program with P.S.D. quadratic matrix.

**Lemma B.1** (Gradient descent for quadratic program). *Let $\boldsymbol{Q} \in \mathbb{R}^{d \times d}$ be a positive semi-definite matrix and $\boldsymbol{p} \in \mathbb{R}^d$ such that $\boldsymbol{p} \in Range(\boldsymbol{Q})$. Then gradient descent initialized at $\boldsymbol{x}_0$ with learning rate $\gamma < \lambda_{\max}(\boldsymbol{Q})^{-1}$ applied to quadratic program*

$$\min_{\boldsymbol{x}} \frac{1}{2} \boldsymbol{x}^\top \boldsymbol{Q} \boldsymbol{x} - \boldsymbol{p}^\top \boldsymbol{x} \tag{9}$$

*converges. Moreover, suppose $rank(\boldsymbol{Q}) = r$. Since $\boldsymbol{Q}$ is P.S.D., let $\boldsymbol{Q} = \boldsymbol{V}\boldsymbol{\Lambda}\boldsymbol{V}^\top$ be the eigenvalue decomposition of $\boldsymbol{Q}$ where $\boldsymbol{V}$ spans the subspace on which $\boldsymbol{Q}$ has strictly positive eigenvalue. Then, the gradient descent converges to*

$$\boldsymbol{x}_t \to \boldsymbol{x}_\infty = (\boldsymbol{I} - \boldsymbol{V}\boldsymbol{V}^\top)\boldsymbol{x}_0 + \boldsymbol{V}\boldsymbol{\Lambda}^{-1}\boldsymbol{V}^\top \boldsymbol{p}.$$

*Proof.* Let $\boldsymbol{V}_\perp \in \mathbb{R}^{d \times (d-r)}$ be the matrix formed by orthonormal basis in the complement of $\text{Range}(\boldsymbol{V})$. As a result, $[\boldsymbol{V}|\boldsymbol{V}_\perp] \in \mathbb{R}^{d \times d}$ is an orthogonal matrix. Let $\boldsymbol{\Lambda} = \text{diag}(\lambda_1, \ldots, \lambda_r)$, where each $\lambda_i > 0$ is strictly positive. Denote the objective function by $f(\boldsymbol{x})$, then $\nabla f(\boldsymbol{x}) = \boldsymbol{Q}\boldsymbol{x} - \boldsymbol{p}$. The gradient descent update is given by

$$\boldsymbol{x}_{t+1} = \boldsymbol{x}_t - \gamma \nabla f(\boldsymbol{x}_t) = \boldsymbol{x}_t - \gamma(\boldsymbol{Q}\boldsymbol{x}_t - \boldsymbol{p}).$$

Next, we decompose $\boldsymbol{x}_t$ into two parts, $\boldsymbol{V}^\top \boldsymbol{x}_t$ corresponding to the range of $\boldsymbol{Q}$ and $\boldsymbol{V}_\perp^\top \boldsymbol{x}_t$ corresponding to the null space of $\boldsymbol{Q}$. To analyze the later part, notice that our assumption $\boldsymbol{p} \in \text{Range}(\boldsymbol{Q})$ implies that $\boldsymbol{V}_\perp^\top \boldsymbol{p} = 0$. Therefore,

$$\boldsymbol{V}_\perp^\top \boldsymbol{x}_{t+1} = \boldsymbol{V}_\perp^\top \boldsymbol{x}_t - \gamma(\boldsymbol{V}_\perp^\top \boldsymbol{Q}\boldsymbol{x}_t - \boldsymbol{V}_\perp^\top \boldsymbol{p}) = \boldsymbol{V}_\perp^\top \boldsymbol{x}_t. = \boldsymbol{V}_\perp^\top \boldsymbol{x}_0.$$

This means that the component of $\boldsymbol{x}_t$ in the null space of $\boldsymbol{Q}$ remains unchanged during the gradient descent update. Now, for the former part,

$$\boldsymbol{V}^\top \boldsymbol{x}_{t+1} = \boldsymbol{V}^\top \boldsymbol{x}_t - \gamma(\boldsymbol{\Lambda}\boldsymbol{V}^\top \boldsymbol{x}_t - \boldsymbol{V}^\top \boldsymbol{p}) = (\boldsymbol{I} - \gamma\boldsymbol{\Lambda})(\boldsymbol{V}^\top \boldsymbol{x}_t) + \gamma\boldsymbol{V}^\top \boldsymbol{p}.$$

Note that $\|\boldsymbol{I} - \gamma\boldsymbol{\Lambda}\|_{op} < 1$ since $\gamma < \lambda_{\max}(\boldsymbol{Q})^{-1}$. Therefore, $\boldsymbol{V}^\top \boldsymbol{x}_t$ is a Cauchy sequence and hence converges. Denote the limit by $\boldsymbol{V}^\top \boldsymbol{x}_\infty$, we have

$$\boldsymbol{V}^\top \boldsymbol{x}_\infty = (\boldsymbol{I} - \gamma\boldsymbol{\Lambda})\boldsymbol{V}^\top \boldsymbol{x}_\infty + \gamma\boldsymbol{V}^\top \boldsymbol{p} \Rightarrow \gamma\boldsymbol{\Lambda}\boldsymbol{V}^\top \boldsymbol{x}_\infty = \gamma\boldsymbol{V}^\top \boldsymbol{p} \Rightarrow \boldsymbol{V}^\top \boldsymbol{x}_\infty = \boldsymbol{\Lambda}^{-1}\boldsymbol{V}^\top \boldsymbol{p}.$$

To summarize, we have $\boldsymbol{V}_\perp \boldsymbol{V}_\perp^\top \boldsymbol{x}_\infty = \boldsymbol{V}_\perp \boldsymbol{V}_\perp^\top \boldsymbol{x}_0$, and $\boldsymbol{V}\boldsymbol{V}^\top \boldsymbol{x}_\infty = \boldsymbol{V}\boldsymbol{\Lambda}^{-1}\boldsymbol{V}^\top \boldsymbol{p}$. More compactly, $\boldsymbol{x}_\infty = (\boldsymbol{I} - \boldsymbol{V}\boldsymbol{V}^\top)\boldsymbol{x}_0 + \boldsymbol{V}\boldsymbol{\Lambda}^{-1}\boldsymbol{V}^\top \boldsymbol{p}.$ $\square$

### B.1 PROOF OF THEOREM 1

We start by recapping the statement of the theorem:

Suppose that $\boldsymbol{X}$ has strictly positive singular values and there exists $\boldsymbol{\theta}$ such that $\boldsymbol{X}\boldsymbol{\theta} = \boldsymbol{y}$. For $\lambda > 0$, gradient descent applied to objectives (6), (7), (8) with learning rate less than $\min\{\sigma_{\max}^{-2}, (\lambda + \sigma_{\max}^2)^{-1}, (\lambda\sigma_{\max}^2 + \sigma_{\max}^2)^{-1}\}$ converges. Moreover, if gradient descent is initialized at the pretrained weights $\boldsymbol{\theta}_0$, they converge to the following solution

$$\textbf{Direct tuning:} \quad \boldsymbol{\theta}_{\text{Direct}} = \left[\boldsymbol{I} - \boldsymbol{X}^\mathsf{T}(\boldsymbol{X}\boldsymbol{X}^\mathsf{T})^{-1}\boldsymbol{X}\right]\boldsymbol{\theta}_0 + \boldsymbol{X}^\mathsf{T}(\boldsymbol{X}\boldsymbol{X})^{-1}\boldsymbol{y},$$

$$L_2 \textbf{ regularization:} \quad \boldsymbol{\theta}_{\text{L2}} = (\boldsymbol{X}^\mathsf{T}\boldsymbol{X} + \lambda\boldsymbol{I})^{-1}\boldsymbol{X}^\mathsf{T}\boldsymbol{y} + \lambda(\boldsymbol{X}^\mathsf{T}\boldsymbol{X} + \lambda\boldsymbol{I})^{-1}\boldsymbol{\theta}_0,$$

$$\textbf{Label annealing:} \quad \boldsymbol{\theta}_{\text{LA}} = \left[\boldsymbol{I} - (1+\lambda)^{-1}\boldsymbol{X}^\mathsf{T}(\boldsymbol{X}\boldsymbol{X}^\mathsf{T})^{-1}\boldsymbol{X}\right]\boldsymbol{\theta}_0 + (1+\lambda)^{-1}\boldsymbol{X}^\mathsf{T}(\boldsymbol{X}\boldsymbol{X})^{-1}\boldsymbol{y}.$$

We will analyze them one by one below.

**Direct tuning.** The objective function for direct finetuning is

$$\min_{\boldsymbol{\theta}\in\mathbb{R}^d} \frac{1}{2}\|\boldsymbol{X}\boldsymbol{\theta} - \boldsymbol{y}\|_2^2.$$

It corresponds to objective in (9) with $\boldsymbol{Q} = \boldsymbol{X}^\mathsf{T}\boldsymbol{X} \in \mathbb{R}^{d\times d}$ and $\boldsymbol{p} = \boldsymbol{X}^\mathsf{T}\boldsymbol{y}$. Since we assume $\boldsymbol{X}$ has strictly positive singular values, let $\boldsymbol{X} = \boldsymbol{U}\boldsymbol{\Sigma}\boldsymbol{V}^\mathsf{T}$ be the compact SVD of $\boldsymbol{X}$, where $\boldsymbol{\Sigma} \in \mathbb{R}^{d\times d}$. Then $\boldsymbol{Q} = \boldsymbol{V}\boldsymbol{\Sigma}^2\boldsymbol{V}^\mathsf{T}$. Let $\boldsymbol{\Lambda} = \boldsymbol{\Sigma}^2$. Then

$$\boldsymbol{V}\boldsymbol{V}^\mathsf{T} = \boldsymbol{X}^\mathsf{T}(\boldsymbol{X}\boldsymbol{X}^\mathsf{T})^{-1}\boldsymbol{X}$$

and

$$\boldsymbol{V}\boldsymbol{\Lambda}^{-1}\boldsymbol{V}^\mathsf{T} = \boldsymbol{X}^\mathsf{T}(\boldsymbol{X}\boldsymbol{X}^\mathsf{T})^{-2}\boldsymbol{X}.$$

Plugging in them into Lemma B.1, we get the desired $\boldsymbol{\theta}_{\text{Direct}}$.

**Label annealing.** Recall the label annealing objective is

$$\min_{\boldsymbol{\theta}\in\mathbb{R}^d} \frac{1}{2}\|\boldsymbol{X}\boldsymbol{\theta} - \boldsymbol{y}\|_2^2 + \frac{\lambda}{2}\|\boldsymbol{X}\boldsymbol{\theta} - \boldsymbol{X}\boldsymbol{\theta}_0\|_2^2.$$

It follows the same story by applying Lemma 9 but with $\boldsymbol{Q} = (1 + \lambda)\boldsymbol{X}^\mathsf{T}\boldsymbol{X}$ and $\boldsymbol{p} = \boldsymbol{X}^\mathsf{T}\boldsymbol{y} + \lambda\boldsymbol{X}^\mathsf{T}\boldsymbol{X}\boldsymbol{\theta}_0$.

$L_2$ **regularization.** With $L_2$ regularization, the objective becomes strongly convex, as a result, the solution does not depend on initialization anymore, as long as the gradient descent converge. In this case, the solution is simply $\boldsymbol{Q}^{-1}\boldsymbol{p}$ with $\boldsymbol{Q} = (\boldsymbol{X}^\mathsf{T}\boldsymbol{X} + \lambda\boldsymbol{I})$ and $\boldsymbol{p} = \boldsymbol{X}^\mathsf{T}\boldsymbol{y} + \lambda\boldsymbol{\theta}_0$. This completes the proof.

