# OpenReview forum: "Memory retaining finetuning via distillation"
_ICLR.cc/2025/Conference — Submitted to ICLR 2025_

### Official Review · Reviewer_TUzD · 2024-11-02

**Soundness:** 4
**Presentation:** 3
**Contribution:** 3
**Rating:** 6
**Confidence:** 3

**Summary:**

The paper provides a solution to the forgetting of pre-training knowledge assuming that pre-training data is unavailable by using label annealing. This method adds a KL divergence term to the loss and regularizes the divergence of the fine-tuning model's predictions to those of the initial model. There are both empirical examples when fine-tuning on different domains, like math and code, instruction fine-tuning, as well as a mathematical intuition for why the solution mitigates forgetting.

**Strengths:**

-The paper offers a novel method to mitigate forgetting using regularization.

-The paper does a great job of motivating the problem and showing how direct fine-tuning leads to forgetting, as well as shows both empirical and mathematical motivation.

-It was really good to compare how this method works both for scenarios where knowledge might be repeated as well as where knowledge is less likely to be repeated, and thus fine-tuning cannot rely on repeated pre-training data.

**Weaknesses:**

-Section 3.2: The method is compared to L2-regression, but for a complete comparison, there should be other commonly used penalty based methods (e.g. EWC (Kirkpatrick et al. 2017)).

-Section 3.3: There are no baselines apart from direct fine-tuning. It would be especially motivating to add other penalty based methods where the hyperparameter can be altered to show the same type of curve and offer direct comparison.

-The method requires loading 2 models into memory at once for each step (the initial model and fine-tuning model). As commonly used LLMs get larger for many language tasks, scaling this may become impractical.

-Table 3: It would be helpful to add the results of only using label annealing for direct comparison

**Questions:**

-It is surprising that L2 regularization does not maintain past task performance with optimal hyperparameter selection. In Table 1, does changing the hyperparameter on the regularization term not allow the model to retain past task information better?

-It is interesting that in Table 3, using replay improves math performance even more so than direct fine-tuning. Does this imply that math data was perhaps part of the pre-training, and thus replaying it allows the model to do better on the current math fine-tuning task?

---

> ### Author Response · Authors · 2024-11-23
>
> We thank the reviewer for their positive assessment and constructive feedback. We address each point below:
>
> Regarding the comparison with other penalty-based methods: We agree that comparing with methods like EWC would strengthen the paper. Our focus was on demonstrating that optimizing a data-dependent objective (e.g., KL divergence) outperforms data-independent penalties (e.g., L2).
>
>  On Section 3.3 baselines: We do have the L2 penalty-based baseline in the paper section. We find that on L2 regularization, even with very large regularization, the model can’t recover the same behavior as the initial mode.
>
>  Regarding memory requirements: The additional memory from loading a frozen copy is modest, since we don't need to store gradients or optimizer states for the frozen model. The full backward and forward pass on a neural network requires 4 times the memory needed for one network weights (1 for forward pass, 1 for gradient, and 2 Adam states). Therefore, using a frozen model doesn’t increase memory requirement by 2 times, but 1.25 times.
>
>  To address the specific questions:
>
> * L2 regularization and past performance: Even with optimal hyperparameter selection, L2 regularization provides limited benefit because it cannot consider how the fine-tuning data interacts with the model's existing knowledge. This is demonstrated in both Table 1 and 2, where L2 regularization fails to prevent significant drops in source benchmarks.
> * Math performance with replay: Yes, we believe this is indeed the case. The improved math performance with replay suggests that some mathematical knowledge exists in the pretraining corpus.

---

> > ### Comment · Reviewer_TUzD · 2024-11-28
> >
> > Thank you for the response. I will maintain my score.

---

### Official Review · Reviewer_Qr1v · 2024-11-03

**Soundness:** 2
**Presentation:** 1
**Contribution:** 2
**Rating:** 3
**Confidence:** 4

**Summary:**

The paper addresses the issue of catastrophic forgetting in large models during finetuning, specifically when capabilities such as in-context learning are lost. The authors propose a method called “label annealing” to mitigate this issue without needing access to the original pre-training data. This is achieved by incorporating a KL divergence term in the loss function to keep the finetuned model close to the pre-trained model. It would be valuable to discuss how this method compares directly to simpler approaches, such as parameter merging.

**Strengths:**

- The problem of knowledge forgetting during instruction finetuning is important.
- The paper provides solid theoretical analysis.

**Weaknesses:**

- The paper is difficult to read due to numerous grammatical errors and typos. These issues detract from the overall clarity and academic tone. For example:
	- Line 133: “more effectove” should be corrected to “more effective.”
	- Line 136: The phrase “Label smoothing is similar to our proposal is that…” should be revised for clarity, perhaps to “… is similar to our proposal in that…”
	- Several sentences, such as those on Lines 167, 175, 202, 205, and 215, are unclear or poorly constructed. These should be reorganized into concise, well-structured parts.
- The proposed use of KL divergence is not particularly novel, as similar techniques have been widely used in instruction finetuning [1] and model alignment (e.g., DPO). More differentiation from existing methods is needed.
- The rationale behind addressing forgetting issues by potentially mixing pre-training and instruction data is not well justified. In real-world implementations like Alpaca, UltraChat, Wizard, and OpenChat, researchers tend to focus on building more diverse instruction datasets rather than revisiting pre-training data. There are also many works on mixing general and specific instruction for instruction tuning [2-3].
- The paper does not provide sufficient examples to demonstrate the benefits of retaining general knowledge obtained during pre-training. This is especially problematic given that retaining knowledge could lead to conflicts and potentially degrade performance on target-domain tasks, as seen in results like Table 2. Additionally, while the concept of alignment tax is mentioned, the paper does not address the potential impact on performance for source tasks like MMLU and TriviaQA during instruction tuning.
- There is no comparison with model merging techniques such as ExPO [4], which would be relevant for a comprehensive evaluation of the proposed approach.

[1] Shi, Zhengyan, et al. "Instruction Tuning With Loss Over Instructions." arXiv preprint arXiv:2405.14394 (2024).

[2] Yuan, Lifan, et al. "Advancing llm reasoning generalists with preference trees." arXiv preprint arXiv:2404.02078 (2024).

[3] Zhang, Kaiyan, et al. "Ultramedical: Building specialized generalists in biomedicine." arXiv preprint arXiv:2406.03949 (2024).

[4] Zheng, Chujie, et al. "Weak-to-strong extrapolation expedites alignment." arXiv preprint arXiv:2404.16792 (2024).

**Questions:**

- Since the pre-training data for LLaMA is not publicly available, have the authors considered using open-source datasets like fine-web [1] as a substitute? It would be helpful to know if the proposed method outperforms simply using pre-training data directly.
 - The claim in Line 235 that “a fixed set of training hyperparameters” is used seems problematic. Instruction tuning is sensitive to hyperparameter choices, and results should ideally be supported by a thorough grid search to ensure reliability.
- The additional KL loss term likely increases the overall training cost. The authors should address how this cost is managed or justified.
- What is the size of the data generated for math/code finetuning? It’s important to consider whether scaling the instruction size could negate the differences between finetuning strategies.
- The method appears less effective than direct finetuning for tasks like HumanEval. What specific advantages does the label annealing method offer for domains like math, if performance gains in target domains are more critical than knowledge retention?

[1] Penedo, Guilherme, et al. "The fineweb datasets: Decanting the web for the finest text data at scale." arXiv preprint arXiv:2406.17557 (2024).

---

> ### Author Response · Authors · 2024-11-23
>
> We acknowledge the writing issues identified and have corrected the grammatical errors and unclear sentences in the lines mentioned. We have updated the submission file with fixed typos and improved presentation.
>
>  Regarding technical concerns:
>
> * Use of KL divergence: We agree that applying KL divergence to language modeling is not novel. Our contribution lies in demonstrating that this technique can effectively prevent forgetting during practical large language model finetuning, supported by comprehensive empirical results across different scenarios.
> * Model merging comparison: Thank you for suggesting the ExPO reference. We will include this discussion in related work. Our paper focuses specifically on methods that only require access to model weights, without additional information available.
> * Benefits of retaining general knowledge: The importance of preventing forgetting is demonstrated clearly in our empirical results. For example, direct finetuning leads to significant drops in general capabilities (e.g., 14.19% drop in TriviaQA). When finetuning a language model for specific capabilities, we don't want it to forget its basic abilities that make it useful as a general-purpose model.
> * Fixed hyperparameters and instruction tuning: Our instruction tuning setup achieves substantial improvements on standard benchmarks \- improving AlpacaEval from 0% to 10%, surpassing the performance of Llama2-chat. This validates our hyperparameter choices.
> * Computational overhead: The additional KL loss term increases training time from 21s to 28s per batch on an 8xH100 node for all the experiments mentioned in the paper. The overhead is modest because the frozen copy of initial weights only requires forward passes, not backward passes.
>
> Regarding your specific questions:
>
> * Data sizes: As mentioned in Section 3.2, we use 179M tokens for math finetuning and 30M tokens for code finetuning.
> * Open-source pre-training data: We explored this direction using RedPajama data, with results reported in the limitations section.
> * HumanEval performance: Label annealing intentionally introduces a tradeoff between downstream and existing capabilities, allowing practitioners to select their preferred operating point based on application needs.

---

> > ### Comment · Reviewer_Qr1v · 2024-11-25
> >
> > Thank you for your response. Regarding model merging, I believe it serves as a strong baseline that should be considered for comparison. This approach can be implemented with access to weights alone, not just ExPO.
> >
> > Additionally, the motivation for balancing general and specific abilities lacks clarity and robustness. The results presented do not convincingly demonstrate an advantage in improving specific tasks while preserving general knowledge.
> >
> > It’s also not accurate to assert a drop in general capabilities without broader validation across more tasks and algorithms. Even when using the same data, methods like DPO and SimPO can yield different outcomes for general abilities.
> >
> > Lastly, there should be specific examples that clearly illustrate the advantages of a specialized generalist approach (retaining general knowledge while enhancing specialized abilities) for particular tasks. For instance, is there a case in math or coding where retaining general knowledge is essential to solving the problem?
> >
> > Considering these concerns, I believe the paper requires further revisions to clarify its motivations and provide additional experiments to validate its effectiveness. Therefore, I choose to maintain my current rating.

---

### Official Review · Reviewer_BE4W · 2024-11-04

**Soundness:** 3
**Presentation:** 3
**Contribution:** 3
**Rating:** 5
**Confidence:** 3

**Summary:**

This paper introduces the idea of label annealing, which is designed to reduce the problem of forgetting knowledge that was learned during pretraining, as part of finetuning.  This is specifically done without having access to the original pretraining data, as is common in a lot of modern practice, e.g. with the LLaMa open-weight models and that precludes direct application of techniques such as experience replay.  The idea here is to keep an independent frozen version of the pretrained model and penalize the finetuning with a relative entropy term between that and the finetuned model with respect to predicted token probabilities.  The validity of this approach is demonstrated through fairly extensive experiments.

**Strengths:**

The paper is motivated by the very practical concern of preventing forgetting as part of the finetuning process in a sociotechnical setting where the pretraining data is not known (or perhaps can only be approximated as in the RedPajama dataset).  As such, any solution in this direction is useful.

The approach is straightforward, backed by basic theoretical explanation, and directly implementable.

Sections 1 and 2.2 are quite clearly written.

**Weaknesses:**

The experimental section is extensive, but also sometimes hard to understand what exactly is demonstrated by the results.  Indeed, in many of the tables, the finetuning doesn't seem to help in advanced benchmarks such as in math.

**Questions:**

What, specifically, are the advantages of the proposed technique that the experiments show?  Is it the ability to have a smooth tradeoff curve between the pretrained and finetuned settings?

---

> ### Author Response · Authors · 2024-11-23
>
> We thank the reviewer for their detailed assessment and constructive feedback. In response to the reviewer's specific comments:
>
>  What are the specific advantages demonstrated in the experiments, and do they primarily show smooth tradeoff capability? The experiments demonstrate two distinct benefits of label annealing:
>
> * In scenarios where forgetting can be mitigated without compromising target performance (Tables 1 and 2), label annealing preserves improvements while preventing knowledge loss. For example, in math finetuning (Table 1), label annealing maintains the improvement in MATH (17.94%) and GSM8K (61.78%) while mitigating the drop in pretraining metrics (MMLU and TriviaQA) that occurs with direct finetuning. Similarly, for code finetuning (Table 2), label annealing preserves most of the HumanEval gains while preventing the dramatic forgetting in mathematics benchmarks that occurs with direct finetuning (MATH drops by 14.73% with direct finetuning).
> * When there are inherent conflicts between pre-training and fine-tuning objectives (Figures 2 and 3), label annealing introduces a smooth tradeoff between the two domains, allowing practitioners to select their preferred operating point.
>
> Regarding limited improvement in advanced benchmarks: The improvements we observe in mathematics benchmarks (e.g., \+2.02% on MATH, \+10.61% on GSM8K) come from fine-tuning on mathematics-related text, not from direct supervised learning on the benchmark training sets. This is an important distinction, as our method improves generalization through domain adaptation rather than through direct task optimization, which typically yields larger but potentially less generalizable gains.
>
> We acknowledge that the presentation of experimental results could be clearer. In our revision, we will highlight the key takeaways more explicitly around each set of results to better guide readers through the demonstrated benefits.

---

### Meta-Review · Area_Chair_VnQw · 2024-12-22

**Metareview:**

The paper tackles the challenge of catastrophic forgetting in large models during fine-tuning, particularly the loss of capabilities such as in-context learning. To address this, the authors propose a method called “label annealing,” which incorporates a KL divergence term in the loss function to maintain the fine-tuned model's proximity to the pre-trained model, all without requiring access to the original pre-training data.

The strengths of the paper lie in its focus on an important research question and its solid theoretical analysis. However, the work has significant weaknesses, including the lack of robust comparisons with key baselines and shortcomings in presentation quality.

In light of these limitations, I recommend rejecting this submission.

**Additional Comments On Reviewer Discussion:**

During the discussion period, Reviewers Qr1v and TUzD actively engaged with the authors.

After carefully reviewing all the concerns raised by the reviewers, I found that the authors failed to adequately address two critical issues: (1) a lack of empirical comparison with key baselines and (2) poor presentation quality.

A major concern raised by all reviewers is the lack of valid comparisons with key baselines, such as model merging. Model merging, which relies solely on leveraging weights, serves as a strong baseline that effectively balances generalizability and task specificity. Unfortunately, the authors did not provide a substantial response to this important concern.

Additionally, the poor presentation quality of the submission was noted by multiple reviewers. Despite receiving this feedback during the rebuttal phase, the authors did not make any significant adjustments to improve the clarity or quality of their work.

These unresolved issues severely limit the validity and potential impact of the paper, reducing its appeal to the broader research community. Consequently, I believe the submission does not meet the high standards expected for acceptance at the prestigious ICLR conference.

---

### Decision · Program_Chairs · 2025-01-22

Reject